# Vaso-Occlusion in Sickle Cell Disease: Is Autonomic Dysregulation of the Microvasculature the Trigger?

**DOI:** 10.3390/jcm8101690

**Published:** 2019-10-15

**Authors:** Saranya Veluswamy, Payal Shah, Christopher C. Denton, Patjanaporn Chalacheva, Michael C. K. Khoo, Thomas D. Coates

**Affiliations:** 1Hematology Section, Children’s Center for Cancer and Blood Diseases, Children’s Hospital Los Angeles, 4650 Sunset boulevard, Los Angeles, CA 90027, USA; sveluswamy@chla.usc.edu (S.V.); pmshah@chla.usc.edu (P.S.); chdenton@chla.usc.edu (C.C.D.); 2Department of Biomedical Engineering, University of Southern California, 1042 Downey Way, Los Angeles, CA 90089, USA; pchalach@andrew.cmu.edu (P.C.);

**Keywords:** sickle cell disease, autonomic nervous system dysfunction, vaso-occlusive crisis, pain, microvascular blood flow

## Abstract

Sickle cell disease (SCD) is an inherited hemoglobinopathy characterized by polymerization of hemoglobin S upon deoxygenation that results in the formation of rigid sickled-shaped red blood cells that can occlude the microvasculature, which leads to sudden onsets of pain. The severity of vaso-occlusive crises (VOC) is quite variable among patients, which is not fully explained by their genetic and biological profiles. The mechanism that initiates the transition from steady state to VOC remains unknown, as is the role of clinically reported triggers such as stress, cold and pain. The rate of hemoglobin S polymerization after deoxygenation is an important determinant of vaso-occlusion. Similarly, the microvascular blood flow rate plays a critical role as fast-moving red blood cells are better able to escape the microvasculature before polymerization of deoxy-hemoglobin S causes the red cells to become rigid and lodge in small vessels. The role of the autonomic nervous system (ANS) activity in VOC initiation and propagation has been underestimated considering that the ANS is the major regulator of microvascular blood flow and that most triggers of VOC can alter the autonomic balance. Here, we will briefly review the evidence supporting the presence of ANS dysfunction in SCD, its implications in the onset of VOC, and how differences in autonomic vasoreactivity might potentially contribute to variability in VOC severity.

## 1. Introduction

Sickle cell disease (SCD) is a monogenic disorder in which a single amino acid substitution in the beta globin gene (A→T) gives rise to the formation of abnormal hemoglobin, i.e., hemoglobin S (HbS) that polymerizes upon deoxygenation [1,2,3]. The long polymers of deoxygenated HbS distort the red blood cell (RBC) into the characteristic sickle shape, a process that is exquisitely sensitive to the concentration of HbS within the cell [4]. “Sickling” is usually reversible once the hemoglobin is reoxygenated in the lungs; however, repeated cycles of deoxygenation/oxygenation result in RBCs becoming irreversibly sickled [5,6]. Sickled RBCs have greatly reduced deformability, becoming rigid, which hampers their ability to navigate capillaries and very small blood vessels, leading to occlusion of the microvasculature with subsequent pain, ischemia and organ damage [3,7]. The pathology of SCD can be directly attributed to HbS polymerization and vaso-occlusion, as well as to secondary effects of chronic hemolysis. In the past few decades, significant advances have been made in our understanding of the complex biochemical and vascular processes that interact with each other to result in vaso-occlusion. Intravascular hemolysis and release of free hemoglobin leading to nitric oxide depletion, endothelial dysfunction, and neutrophil and platelet adhesion are all playing an important contributory role to the pathophysiology of SCD, and have been extensively discussed elsewhere [7,8,9].

The hallmark of SCD is the sudden onset of episodes of excruciating musculoskeletal pain known as vaso-occlusive crises (VOC). VOC frequency is tremendously variable among patients, even if the SCD genotype (SS, SC, etc.) is the same. A small subset of patients have more frequent and more severe VOC, accounting for most hospitalizations for pain [10], while a substantial portion of patients have a relatively low frequency of VOC pain events. Genetic modifiers of disease such as co-inheritance of alpha thalassemia, beta globin haplotypes and elevated hemoglobin F are thought to act in concert with various mediators of hemolysis and inflammation to modulate disease severity [11,12]. However, these biological and cellular profiles still do not fully account for disease variability, which remains mostly unexplained. Autonomic dysfunction in SCD subjects has been described as related to disease severity. However, its role in initiating and promoting sickle vaso-occlusions has not been extensively considered.

Based on clinical histories and anecdotal comments from patients, VOC pain evolves over minutes to hours and can be triggered by stress, cold temperature and pain itself. The mechanism of the transition from steady state to VOC, and the mechanism by which these triggers can initiate the transition are not known. Here, we will review the basic mechanism of sickle vaso-occlusion and discuss the implications of autonomic vascular dysfunction in initiating and maintaining VOC.

## 2. Pathophysiology of Sickle Vaso-Occlusion

Four decades ago, Eaton and Hofrichter drew the connection between regional blood flow, kinetics of hemoglobin polymerization and sickle vaso-occlusion [13,14,15,16]. This model is presented in Figure 1 and is the basis for this discussion about the pathophysiology of SCD. Normal RBCs are extremely deformable and bend as they navigate through the capillaries in the microvasculature to deliver oxygen to the tissues. RBCs carrying oxygenated HbS are also very flexible, but once they deoxygenate in the microvasculature, HbS polymerizes leading to formation of very rigid sickle-shaped RBCs. Rather than being instantaneous, HbS polymerization becomes maximal after a delay time (T**_d_**: Figure 1), which allows most of the RBCs to escape the microvasculature into the larger venules before sickling occurs [17,18]. If the regional blood flow decreases, the transit time (T**_t_**: Figure 1) of the RBCs through the microvasculature is prolonged, and sickling occurs while the RBCs are still in the capillaries, resulting in entrapment. Thus, the likelihood of vaso-occlusion will depend on (1) the delay time between deoxygenation and polymerization of HbS; and (2) the blood flow rate in the microvasculature.

Lengthening the delay time decreases the likelihood of vaso-occlusion. The delay time is dependent on the 30th power of the HbS concentration within the RBC; thus, a very small decrease in intracellular HbS concentration, such as when increasing the HbF percentage, increasing red cell volume or with changes in red cell hydration will cause significant prolongation of the delay time [20]. Many, if not most, of the beneficial effects of treatment with hydroxyurea in SCD can be attributed to its influence on the delay time to polymerization [20]. Co-inherited genetic conditions such as hereditary persistence of HbF and relatively higher HbA1 as in S-β^+^ thalassemia also exert their disease modulatory effect through dilution of intracellular HbS. Several attempts at decreasing HbS polymerization by increasing its oxygen affinity have been proposed [20], and one agent with such activity, Voxelotor, has recently been shown to increase hemoglobin, including HbS, and reduce the hemolytic rate in patients with SCD [21]. On the other hand, erythrocyte dehydration and reduced HbS oxygen affinity significantly shorten the delay time and predispose to vaso-occlusion [3].

The model depicted in Figure 1 highlights the role of microvascular flow rate. Cardinal contributors to SCD pathogenesis such as changes in blood rheology, hemolysis and inflammation influence vaso-occlusion by modulating the microvascular flow rate. Sickled RBCs are less deformable and increase blood viscosity at all shear rates, an effect only partially offset by anemia [5,22]. Elevated hematocrit and white cell count have been shown to independently increase the incidence of pain and acute chest syndrome by mediating blood viscosity and flow in the post–capillary venule [7,10]. Sickled RBCs have a much shorter life span, leading to both extra- and intravascular hemolysis with ensuing anemia. The degree of hemolysis within each SCD genotype can vary, contributing to some of the phenotypic variability of the disease [23]. Cell-free hemoglobin and Arginase-1 that are released during intravascular hemolysis lead to nitric oxide depletion, promote vasoconstriction, endothelial dysfunction and vascular remodeling [9,24]. Indeed, the association of markers of hemolysis with pulmonary hypertension and chronic kidney disease has been well documented over the past few decades [25,26]. Cell-free hemoglobin and heme also promote oxidative stress and membrane damage in both the vasculature and the RBCs, which leads to endothelial activation and increased adhesion of sickled RBCs to the endothelium [3,5,27]. The activated endothelium releases a multitude of inflammatory mediators such as endothelial adhesion molecules, E- and P- selectins that create a pro-inflammatory state and release vaso-active peptides such as Endothelin-1 that promote vasoconstriction [8,28,29]. Activated neutrophils and platelets show increased adhesion to the endothelium and to sickled RBCs, primarily in the post capillary venules, and likely promote vaso-occlusion by decreasing blood flow rate through the microvasculature. Over the past decade, there has been an incredible growth in our understanding of these complex interactions of cellular elements and their clinical implications for sickle vaso-occlusion. Recent trials with antibodies to P-selectin have shown promising results in decreasing the frequency of VOC in adults with SCD [30], thus underlining the significance of inflammatory mediators in sickle vaso-occlusion. In general, all of these factors act to modulate microvascular flow and thus the transit time of erythrocytes in the microvasculature, as recently reviewed in detail [9,22,29].

In our view, the sum of these inflammatory, rheological and biochemical processes produces a microvascular flow equilibrium, with micro-occlusions likely occurring from time to time in the vast capillary networks, which are not clinically perceived by the patient or may only cause transient symptoms that recover quickly. Whether or not a perturbation like transient decrease in regional perfusion results in cascading occlusion of multiple capillary beds with clinically significant vaso-occlusion would depend on the current set point of the equilibrium and the magnitude of the perturbation [19]. Based on the time course of evolution of VOC reported by patients, the triggering event occurs over minutes to hours, and the pain usually starts in one area and moves to other non-contiguous regions. This makes many of the aforementioned biochemical, rheological and inflammatory processes unlikely candidates as triggers. Interestingly, many SCD patients report experiencing an aura that precedes onset of VOC, as if they could somehow sense a change that may upset this equilibrium enough to trigger VOC. The mechanism of this commonly reported phenomenon and its link to VOC is unknown, but it might be that the aura reflects neural-mediated vascular instability that then affects transit time, similar to the vascular instability that precedes migraine [31,32].

## 3. Autonomic Dysfunction and Peripheral Vasoreactivity in SCD

Based on the polymerization of deoxy-HbS, hypoxia has been assumed to be a trigger of VOC and in fact, nighttime hypoxia was shown to predict stroke in children with SCD [33,34]. We simulated nighttime hypoxia by exposing SCD and control subjects to five breaths of 100% nitrogen, expecting to see a drop in microvascular blood flow [35,36]. The experimental hypoxia actually induced dramatic cardiac parasympathetic nervous system withdrawal in SCD subjects, but not in controls (*p* < 0.01) (Figure 2), and we did not observe any changes in blood flow associated with hypoxia. Instead, we saw periodic vasoconstriction events that were almost perfectly aligned with 78% of deep breaths or sighs in SCD patients and only 17% in controls (*p* < 0.001). These vasoconstrictions are neural-mediated responses triggered by stretch receptors in the chest [37]. The fact that a small perturbation like a sigh triggers vasoconstriction in SCD and not controls suggests that ANS is much more responsive in SCD than controls [35,36] and that ANS activity may play a role in SCD pathophysiology. While nighttime hypoxia may predict stroke events, it has not been associated with frequency of VOC [38]. Thus, it is intriguing to speculate whether this neural-mediated hypersensitivity to vasoconstriction in response to sigh is a reflection of a more global increased propensity to vasoconstriction and may play a role in VOC frequency by increasing entrapment of the sickled RBCs. Asthma, which places significant stretch stimulus to the chest wall, is a well-known predictor of VOC and interestingly, while nighttime hypoxia during sleep studies did not predict painful VOC, the apnea-hypopnea index that is associated with chest wall stretch came very close to statistical significance for predicting VOC [38]. These studies support a role for ANS activity in SCD.

Several studies have documented ANS abnormalities in SCD that associate with various outcomes. The ANS is the major regulator of involuntary bodily functions, including cardiac activity, respiration and peripheral vascular function, through complex and recurrent interactions involving central and peripheral neural pathways [39,40]. Assessment of cardiac beat-to-beat variability (HRV) is a well validated, ubiquitous tool used to measure cardiac autonomic activity and provides a window into the general autonomic balance [41,42,43]. High frequency power (HFP) derived from spectral analysis of the cardiac beat-to-beat interval represents parasympathetic activity and low-frequency power (LFP) reflects a combination of sympathetic and parasympathetic activity [41]. Loss of HRV is an independent predictor of mortality in several cardiac disease states [43]. In one of the earliest studies evaluating HRV in SCD, Romero et al. found that 58% of SCD subjects had abnormal cardiac autonomic balance, implying a role for dysautonomia in the high frequency of sudden death in SCD patients [44]. More recently, autonomic activity has been associated with disease severity in SCD [45,46,47]. Reduced parasympathetic activity at baseline has been associated with increased risk of VOC [46] and acute chest syndrome [48]. Pearson et al. showed that children with increased parasympathetic withdrawal during social and emotional challenges were noted to have more severe disease [45], and suggested that altered autonomic tone might exacerbate pain episodes through increased peripheral vasoconstriction. Indeed, SCD patients have been noted to have a sympathetic dominance in their cardiac autonomic activity during VOC compared to steady state [49]. While the cardiac autonomic balance and peripheral vascular autonomic activity are closely linked via complex neurovascular signaling pathways [40,50], they are not a direct reflection of each other. Although these studies associate parasympathetic withdrawal with disease severity, the question remains as to whether the dysautonomia is causative rather than a sequela of severe disease. Interestingly, sickle mice have markedly increased neuronal outflow in response to pain based on direct measures in the spinal cord [51], and patients with sickle cell trait have measurable abnormalities in ANS function [52]. These studies all show associations between ANS dysfunction and SCD, but they do not show any link to the trigger of VOC. Given that SCD is essentially a disease of blood flow, the role of autonomic modulation of blood flow has likely been underestimated thus far in its pathophysiology.

Our initial studies and the literature made us wonder whether regional pain itself could be a trigger of VOC. Motivated by our data that sighs induce vasoconstriction in SCD subjects, we investigated the hypothesis that pain will induce a vasoconstriction response. SCD and control patients were exposed to painful heat stimuli applied on one arm, and bilateral finger blood flow responses were measured by laser-Doppler flow and plethysmography. As displayed in Figure 3, the vasoconstriction response occurred simultaneously in the forehead and both hands quickly after application of the pain stimulus, suggestive of a global autonomic neural-mediated response. The degree of the vasoconstriction response was greater in SCD subjects than in controls, consistent with ANS hypersensitivity [53,54].

There was significant cardiac parasympathetic withdrawal with pain stimuli, further supporting the autonomic involvement. Thus, local pain causing a global vasoconstriction and a decrease in the microvascular flow rate could lead to further vaso-occlusion and pain, creating a positive feedback system that would amplify the vasoconstriction response. Interestingly, when we assessed the microvascular blood flow trends over the entire duration of the pain experiments, some vasoreactivity phenotypes characterizing individual subjects were apparent. For example, in Figure 4, each heat spike within a series causes vasoconstriction and blood flow recovers in between individual pain pulses in Subject 1, whereas Subject 2 has prolonged vasoconstriction that lasts the entire duration of the sequence of short pulses with near-full recovery of flow after the sequence ends. Subject 3 has a progressive decrease in flow throughout the testing period, suggestive of an underlying increased neural/vascular tone. In the few subjects who had a repeat study, these vasoreactivity trends were reproducible. In fact, an autonomic vasoconstriction reactivity signature that is inherent to the individual appears to be present, as further evidenced in our experiments with autonomic tilt table testing [55].

Tilting a human subject from a supine to near-upright position (head-up-tilt: HUT) induces a transient drop in blood pressure that is normally followed by peripheral vasoconstriction and increase in heart rate. Thus, the blood pressure normalizes due to the adaptive autonomic responses. Out of the 66 subjects (27 SCD, 13 anemic and 26 non-anemic controls) who were subjected to HUT testing, only 30% of SCD subjects had a normal response (tachycardia + vasoconstriction; Figure 5) [55]. All subjects, except for one, who responded with only peripheral vasoconstriction had SCD. Baseline parasympathetic activity in the lower 10th percentile resulted in a 70% chance of responding to HUT with vasoconstriction alone, even after adjusting for hemoglobin level [55]. These data strongly suggest that loss of parasympathetic activity results in a vasoconstrictive phenotype, which would favor increased RBC transit time and likelihood of VOC (Figure 5). These data also suggest that subjects might be classified based on vaso-reactivity and ANS features, which could be important to understand and perhaps treat human disease states like SCD [55]. We would predict that SCD individuals like Subject 3 in Figure 4 or in the vasoconstriction-only quadrant of Figure 5 would be more likely to have more frequent VOC.

The detailed view of the vasoconstriction response to thermal pain (Figure 6) shows a small decrease in flow associated with each heat–pain pulse. These small decreases with each heat pulse were statistically significant in 95% of all subjects [54]. However, the most striking finding was the significant vasoconstriction that started before the painful stimulus was applied, when the research assistant told the subject that the pain stimulus was about to happen (Figure 6) [54]. It has been well documented that SCD subjects experience pain-related anxiety and there is good evidence associating anxiety and mental stress to increased crisis frequency and poor quality of life measures in SCD [56,57]. In fact, a bidirectional relationship exists between mental stress and pain of any type, with stress precipitating or augmenting pain by unknown mechanisms [58]. When we subjected our patients to experimental mental stress through standardized mental tasks and to a pain anticipation task (i.e., subjects were warned of upcoming pain, but no pain was applied), we found that the experimental mental stress tasks caused significant vasoconstriction (Figure 7) and parasympathetic withdrawal in both SCD and controls [59]. The pain anticipation task caused the greatest vasoconstriction among all mental stress stimuli, suggesting a potential mechanism by which mental stress and pain-related anxiety could precipitate pain crisis in SCD. We did not detect a difference between SCD and controls in their response to mental stress stimuli, although we did see a significant relation between anxiety scores from standard questionnaires and tonic vasoconstriction in SCD that was not present in controls [59].

The etiology of autonomic dysfunction in SCD remains unclear. Interestingly, Connes et al. showed that individuals with sickle cell trait had abnormal nocturnal cardiac autonomic activity and that there was a correlation between blood viscosity/red cell rigidity and HRV parameters [52,60]. This association between altered blood rheology and decreased HRV has been well documented in cardiac diseases, and it has been postulated that increased blood viscosity interferes with the red cell transit through the micro-capillaries with subsequent decrease in tissue oxygenation and impairment of cardiac activity. This same theory might apply in SCD, especially given the conclusive evidence of myocardial fibrosis and diastolic dysfunction observed in SCD [61]. Furthering the evidence of peripheral autonomic vascular dysfunction in SCD, Esperance et al. showed abnormal peripheral vasoconstriction responses to inspiratory breath hold in SCD [62] and more recently, SCD subjects have been shown to have abnormal activity of sweat glands in their hands and feet, which is a direct measure of peripheral sympathetic activity [63]. Chronic intermittent hypoxia can increase the sensitivity of peripheral and central chemoreceptors leading to increased sympathetic dominance in obstructive sleep apnea and cardiac disease states [64,65,66] and oxidative stress with resultant depletion of nitric oxide and production of endothelin-1 has been implicated in the priming of baroreceptor sensitivity [66]. In fact, there is likely a multitude of factors interacting together to influence the autonomic balance in SCD, and subsequently, the vasoreactivity.

## 4. Conclusions

It is clear that patients with SCD have a hyper-responsive autonomic nervous system. Abnormalities in heart rate variability have been observed in several studies of SCD subjects, though it is only recently that neural-mediated vasoconstriction and dysautonomia have been considered to possibly play a significant role in causing sickle VOC [19,35,53,54,59,67]. The fact that ANS dysfunction is also observed in the sickle trait [52] and that neural pain responses are increased in the sickle mouse [51] raises the interesting thought that HbS might, in some way, be primarily involved in causing this dysautonomia and poses the very intriguing question of why would a single amino acid substitution in the beta chain of hemoglobin affect ANS function? Autonomic modulation of peripheral microvasculature and vascular reactivity provide a plausible additional layer to the already complex biology of sickle vasculopathy [3]. Concurrent vasoconstriction in multiple capillary beds will have a cascading effect to decrease microvascular blood flow and promote vaso-occlusion, thus tipping the balance from steady state to clinically apparent VOC. While the cause for autonomic dysfunction in SCD remains elusive to date, there is strong evidence of cardiac and peripheral vascular dysautonomia in SCD with a subset of patients exhibiting an increased peripheral vasoconstrictive tone. If this logic is correct, SCD subjects with more vasoconstrictive profiles (Figure 4 and Figure 5) would likely have more frequent pain crises. The fact that humans may each have characteristic autonomic vascular reactivity likely contributes to the clinical variability in VOC frequency and is probably also important in other diseases. Furthermore, recent pilot studies have shown that cognitive-based therapies can modulate peripheral vasoconstriction in SCD [68] and might offer a new approach for VOC prevention.

## Figures and Tables

**Figure 1 jcm-08-01690-f001:**
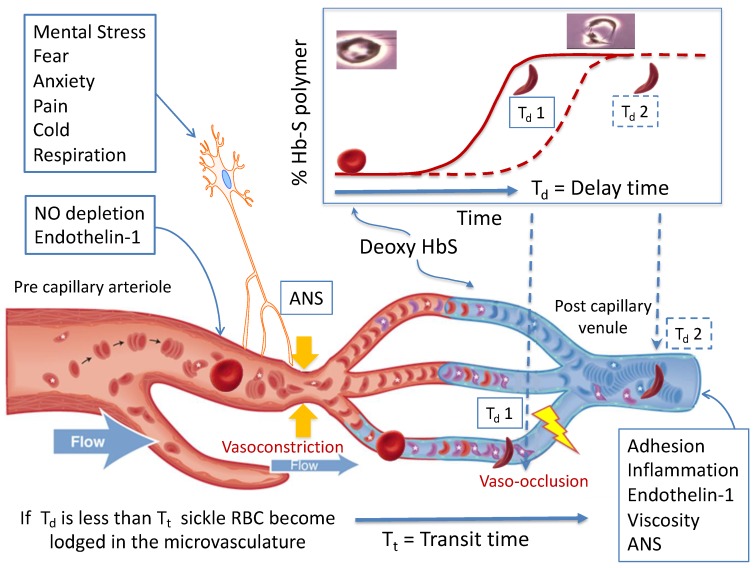
The basic pathophysiological model of sickle vaso-occlusion suggests that microvascular occlusion will occur if the delay time from deoxygenation of HbS to polymerization (T_d_ 1) is shorter than the microvascular transit time (T_t_), as depicted in the lower two vessels. If delay time is longer (T_d_ 2), or if blood is flowing faster, the red cell transition to a rigid shape takes place in a larger diameter post capillary vessel and occlusion is less likely to occur, as depicted in the top micro-vessel. Processes like nitric oxide (NO) depletion and endothelin-1 levels in the precapillary arterioles, and adhesion, inflammation, and viscosity in the post-capillary venule establish a steady state microvascular flow “tone”. Autonomic nervous system (ANS)-mediated vasoconstriction can decrease blood flow within seconds, increasing transit time and the likelihood of entrapment of rigid RBC. Image reprinted with permission from [19].

**Figure 2 jcm-08-01690-f002:**
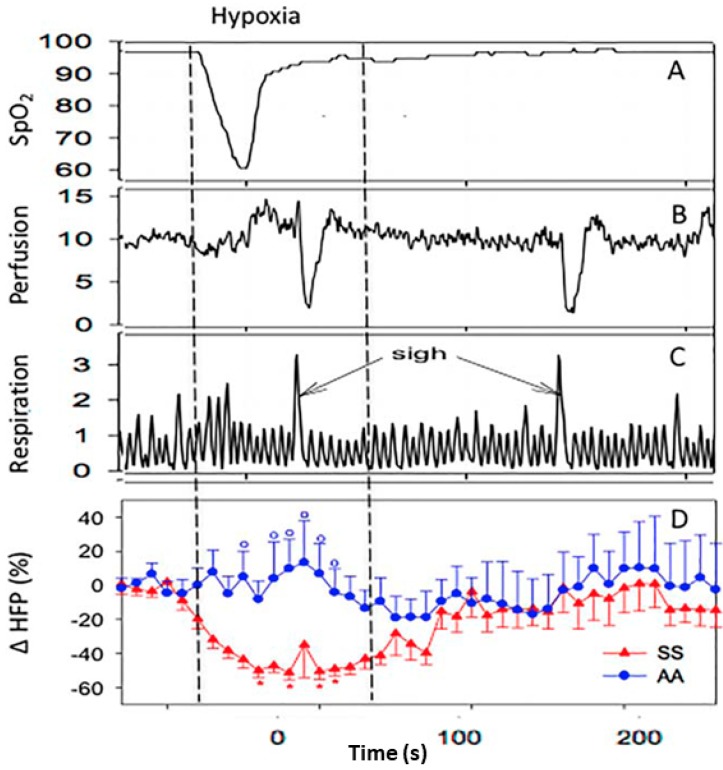
Experimental exposure to five breaths of 100% nitrogen caused desaturation similar to what can happen during sleep. Panels show change in oxygen saturation (**A**), finger blood flow (**B**), respiration (**C**) in a single sickle cell disease (SCD) subject, and change in average parasympathetic activity (cardiac high frequency power; HFP) in 11 SCD and 14 control subjects (**D**). Hypoxia resulted in significant parasympathetic nervous system withdrawal in SCD subjects, but not in controls (**D**). Hypoxia was not associated with a decrease in microvascular perfusion. However, periodic episodes of vasoconstriction (**B**) occurred at about 3.8 s after 78% of sighs (**C**) in SCD subjects versus only 17% in controls (*p* < 0.001). From [36] with permission.

**Figure 3 jcm-08-01690-f003:**
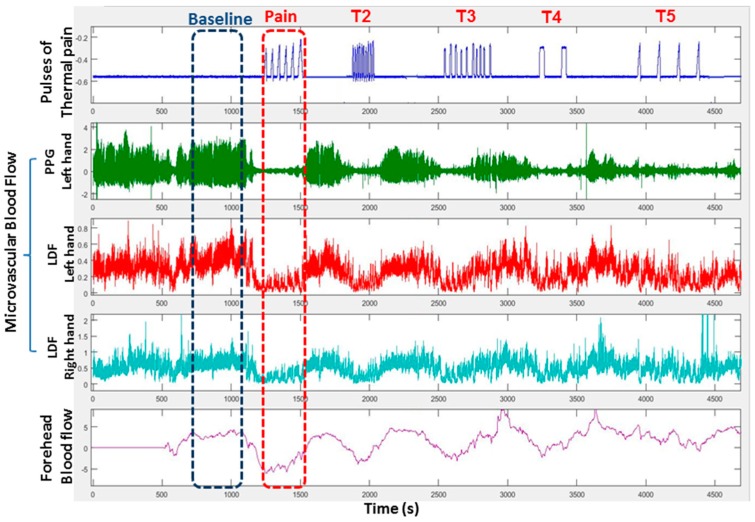
This recording of microvascular perfusion in the right and left index fingers and forehead shows rapid global vasoconstriction in response to each series (Pain, T2–T5) of painful pulses of heat delivered to the right thenar eminence. (PPG = photoplethysmography, LDF = laser doppler flow).

**Figure 4 jcm-08-01690-f004:**
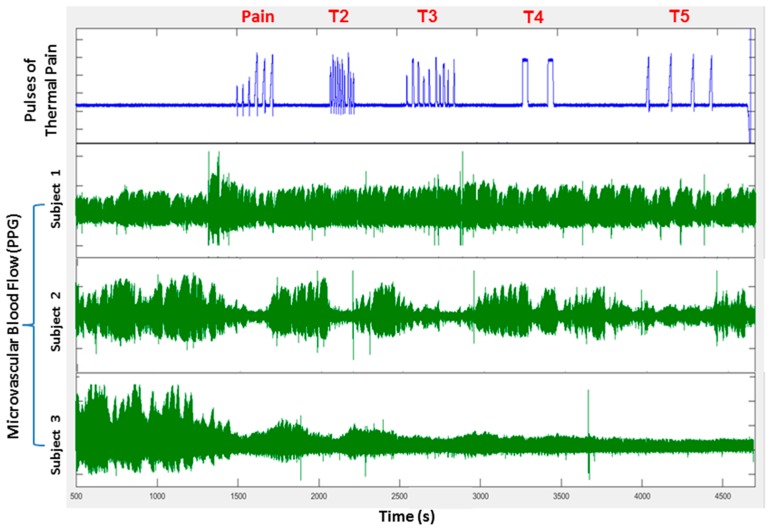
Microvascular blood flow patterns recorded in three subjects in response to several sequences of painful heat pulses (pain, T2–T5) demonstrate significant subject-to-subject variability in neural-mediated vasoconstriction responses. (1) Subject 1 vasoconstricts and recovers blood flow between individual pain pulses; (2) Subject 2 has prolonged vasoconstriction with every sequence of pain stimuli, but recovers blood flow between the sequences; and (3) Subject 3 remains vasoconstricted after the initial stimulus with poor blood flow recovery.

**Figure 5 jcm-08-01690-f005:**
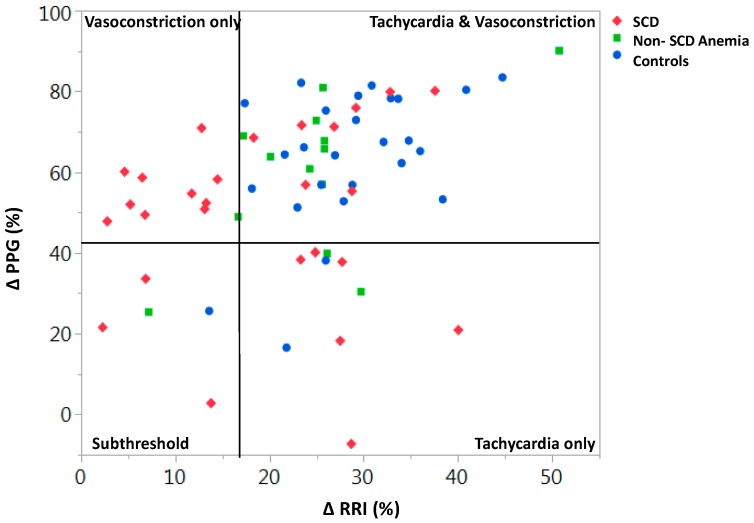
The subjects in the upper-left quadrant have only vasoconstriction in response to head-up-tilt on tilt table testing, in comparison to the normal response of increase in heart rate and vasoconstriction in the upper-right quadrant. The subjects in the vasoconstriction-only group are almost exclusively SCD individuals. Having parasympathetic activity in the lower 10th percentile gives an SCD individual 76% probability of having a vasoconstriction-only phenotype (*p* < 0.01). (After [55] with permission).

**Figure 6 jcm-08-01690-f006:**
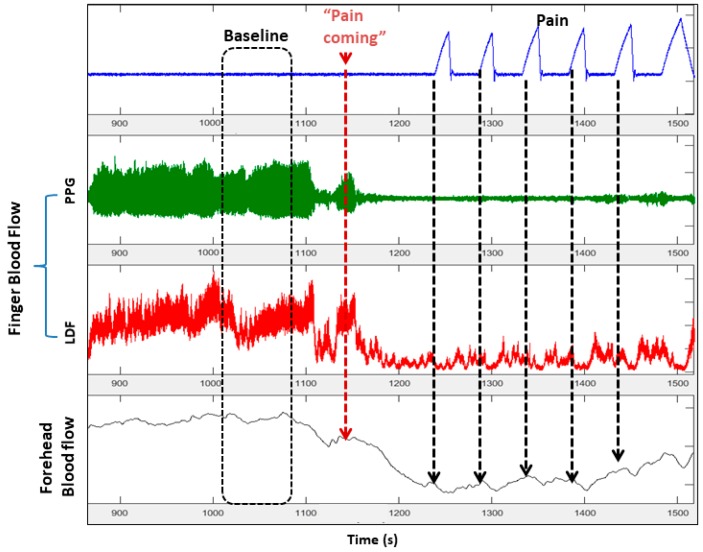
This zoomed section of data shows that significant global vasoconstriction starts when the subjects is informed that “the pain stimulation will start in about two minutes” with additional decreases corresponding to each of the heat pulses. Clearly, anticipation of pain, as well as pain itself, causes vasoconstriction.

**Figure 7 jcm-08-01690-f007:**
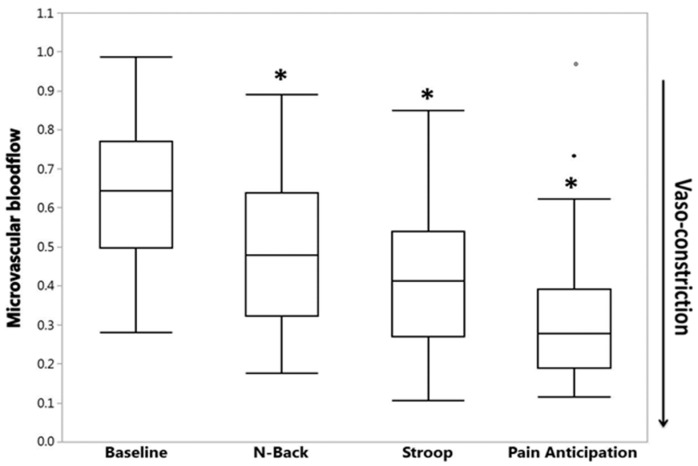
Experimental mental stress causes decrease in microvascular blood flow. However, being told you are going to feel pain is the most powerful stimulus for vasoconstriction (*, *p* < 0.0001 decrease in median blood flow compared to baseline; the two dots represent outliers). (After [59] with permission).

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
