# Peer review of "Vaso-Occlusion in Sickle Cell Disease: Is Autonomic Dysregulation of the Microvasculature the Trigger?"

_jcm, 2019, doi:10.3390/jcm8101690_

Round 1

Reviewer 1 Report

This manuscript summarizes the limited evidence that dysfunction in autonomic nervous system regulation of blood vessel tone may be a trigger (or “the trigger”) for initiation of vaso-occlusive crisis in patients with sickle cell disease. In general, this is a clearly written review manuscript and the perspective is balanced.  I have no major comments.

Minor Comments:

Line 43: “heme” should be changed to “hemoglobin” because chemically, heme will not scavenge NO, but hemoglobin will. Technically, it is oxyhemoglobin that scavenges NO. The authors report empiric data on autonomic instability from their group and others in sickle cell disease and in sickle cell trait. They point out that somehow a beta globin mutation alters autonomic tone and offer some speculation.  They could point out the precedent that alpha globin has been found to be expressed in endothelial cells, where it regulates NO-mediated vascular tone [PMID: 23123858].  It could also be speculated that beta globin hypothetically might be expressed in a cell type that in some manner regulates autonomic tone.  This is quite speculative, but I think OK in a review of this sort.

Author Response

Thank you for the comments

1) As pointed out,we have switched heme to hemoglobin in line 44 as the scavenger of NO in the endothelium

2) We would prefer not to speculate on the mechanism for how the beta globin gene might regulate autonomic tone given lack of any supportive data.

Reviewer 2 Report

This is an excellent and well written review.  

My only comment is that the authors in discussing references 53-55 say that "Subject 3 in Figure 4 or in the vasoconstriction only quadrant of Figure 5 would be more likely to have more frequent VOC".  

Since at least the senior author was an author on references 53-66, can the authors comment on any data they have in this regard or whether they are currently exploring whether this is true?

Thank you

Author Response

Thank you for the comments. We have recently acquired new data that shows that the magnitude of vasoconstriction is a indeed a predictor of VOC and this data will be presented in the upcoming American Society of Hematology meeting.